# The Adaptive Value of Chromosomal Inversions and Climatic Change—Studies on the Natural Populations of *Drosophila subobscura* from the Balkans

**DOI:** 10.3390/insects14070596

**Published:** 2023-07-01

**Authors:** Goran Zivanovic, Concepció Arenas, Francesc Mestres

**Affiliations:** 1Department of Genetics, Institute for Biological Research “Sinisa Stankovic”—National Institute of Republic of Serbia, University of Belgrade, Bulevar Despota Stefana 142, 11000 Belgrade, Serbia; goranziv@ibiss.bg.ac.rs; 2Departament de Genètica, Microbiologia i Estadística, Secció d’Estadística, Universitat de Barcelona, Av. Diagonal 643, 08028 Barcelona, Spain; carenas@ub.edu; 3Departament de Genètica, Microbiologia i Estadística, Secció de Genètica Biomèdica, Evolutiva i Desenvolupament and IRBio (Institut de Recerca per la Biodiversitat), Universitat de Barcelona, Av. Diagonal 643, 08028 Barcelona, Spain

**Keywords:** adaptation, natural selection, global warming, chromosomal thermal index, temperature

## Abstract

**Simple Summary:**

Climatic change is a serious problem because it can irreversibly modify most ecosystems. Many species try to survive by adapting, through natural selection, to the new climatic conditions. In this scenario, the inversion chromosomal polymorphism of *Drosophila subobscura* is an excellent natural system through which to study the adaptations of organisms to climatic change. For analyzing the response of inversions to global warming, the chromosomes of annual samples (2019–2022) from a Petnica (Serbia) *D. subobscura* population were analyzed. These results were compared with those from other populations of this species, but belonging to different climatic regions (Avala, also in Serbia, and Font Groga, Spain). In all cases, significant differences were observed, indicating that the inversion composition conferring adaptive capacity to flies in these regions was different. In Petnica, not all inversions responded in the same way to climatic variables: those located in E and U chromosomes presented more changes. Moreover, by comparing data from 2019–2022 with those from a previous sample also from Petnica (1995), it was possible to relate the changes in meteorological variables to frequency variations of ‘cold’-, ‘warm’-, and ‘non-thermal’-adapted inversions for these years. These results allow for a better understanding of the genetic adaptations to new environmental conditions.

**Abstract:**

The adaptive value of the *Drosophila subobscura* chromosomal inversion polymorphism with regard to environmental effects is well-known. However, the specific details of the inversion adaptations to the global warming scenario deserve to be analyzed. Toward this aim, polymorphism and karyotypes were studied in 574 individuals from Petnica (Serbia) in annual samples taken in June for the period 2019–2022. Comparing the results of Petnica (Cfa: humid subtropical climate) with those from Avala (Serbia: Cfb, temperate oceanic climate) and Font Groga (Barcelona, Spain; Csa: hot-summer Mediterranean climate), significant differences were observed for their chromosomal polymorphism. In Petnica, inversions from U and E chromosomes mainly reacted significantly with regard to temperature, humidity, and rainfall. Moreover, the inversion polymorphism from Petnica (2019–2022) was compared with that from 1995. In this period, a significant increase in mean and maximum temperature was observed. However, to properly explain the observed variations of inversions over time, it was necessary to carefully analyze annual seasonal changes and particular heat wave episodes. Interestingly, yearly fluctuations of U chromosome ‘warm’-adapted inversions corresponded with opposite changes in ‘non-thermal’ inversions. Perhaps these types of inversions were not correctly defined with regard to thermal adaptation, or these fluctuations were also due to adaptations to other physical and/or biological variables. Finally, a joint study of chromosomal inversion polymorphism from many Balkan populations of *D. subobscura* indicated that different climatic regions presented distinct composition, including thermal-adapted inversions.

## 1. Introduction

Global warming on Earth is considered, without any doubt, a fact. There is a large body of evidence corroborating this phenomenon [1,2]. It is a threat to ecosystems that can have serious consequences, including for human beings [3,4,5,6]. Under these circumstances, organisms have the following options: to acclimate (phenotypic plasticity), to adapt through genetic changes, or to migrate to more favorable regions [7]. Not being able to carry out any of these strategies will produce the extinction of the population or the species. Of these, adaptation is considered a cornerstone of Evolutionary Biology [8,9]. As a consequence, a large number of studies have been carried out on the topic of genetic adaptation to global warming, with the *Drosophila* genus being a group of model species. Classically, *Drosophila* chromosomal inversions have been very valuable for understanding the phenomenon of genetic adaptation (for a revision, see [10]). Currently, they are also useful for studying adaptation to global warming [11,12,13,14,15].

For this kind of research, *Drosophila subobscura* is especially interesting due to its rich inversion chromosomal polymorphism [16,17]. This species inhabits vast areas of the Palearctic region. It was detected that many chromosomal inversions are distributed according to latitudinal clines, a result that could be due to adaptation or historical factors [18]. However, the colonization of South [19] and North America [20] by this species clearly demonstrated the adaptive value of chromosomal inversions, because latitudinal clines in the same direction as those observed in the Palearctic region were reported in a short period of time [21]. Thus, the latitudinal clinal variation of certain inversions’ frequencies would likely indicate adaptations with respect to certain climatic and/or biological variables. In this context, the pioneering research of Menozzi and Krimbas [22] reported the relationship between chromosomal inversions and climatic variables, and a few years later Orengo and Prevosti [23] related the temporal changes in polymorphism with temperature. To better understand the relationship between chromosomal inversions and climatic variables, two research approaches were developed for *D. subobscura*: laboratory experiments or the direct study of natural populations. According to the first approach, the variation in frequency for different inversions under different laboratory conditions, thermotolerance and co-adaptation, thermal preferences, and thermotolerance and heat stress resistance were studied [24,25,26,27,28]. On the other hand, there are a large number of studies on *D. subobscura* natural populations analyzing the variation of inversions (composition and frequency) over time and the parallel increase in temperature due to global warming. These variations were studied in the Palearctic populations of the central area of the species distribution [29,30,31,32,33], American colonizing populations [34,35], island-isolated populations [36], and Palearctic populations from marginal areas of *D. subobscura* distribution [37]. However, the adaptation of particular inversions to climatic variables deserves to be studied in depth.

Assuming the hypothesis that chromosomal inversions are adaptive to environmental factors (physical or biological), our main aim was to ascertain the possible relationship between particular inversions and climatic variables. For this purpose, we analyzed the chromosomal composition of *D. subobscura* flies collected in the Serbian population of Petnica. The chromosomal inversion polymorphism of this population was studied using large samples in four consecutive years (2019–2022) during the same season, with the purpose of finding out its possible relationship with some environmental variables (temperature, humidity, and rainfall). According to the Köppen climatic classification [38,39], Petnica belongs to the Cfa type (humid subtropical climate) and for this reason its inversion polymorphism was compared with other previously studied *D. subobscura* populations present in other climates: Avala (Cfb: temperate oceanic climate or subtropical highland climate) also in Serbia [40] and Font Groga (Csa: hot-summer Mediterranean climate) close to Barcelona (Spain) [41]. With this trilogy, we expected to achieve a more detailed view of the adaptive value of inversions with regard to climate variables. Furthermore, with the chromosomal polymorphism data from Petnica obtained in 1995 [42] and 2010 [43], it was possible to ascertain whether this polymorphism had changed over time and in the sense predicted according to global warming. Finally, as chromosomal inversion data were reported in Serbia in the last thirty years [40,42,43,44,45,46], they were revisited to obtain a general overview of the adaptive relationship between inversion polymorphism and environmental variables.

## 2. Materials and Methods

### 2.1. Drosophila subobscura Sampling and Chromosome Preparation 

Individuals were collected from the small village of Petnica (44°14′50′′ N, 19°55′51′′ E), in the district of Kolubara (Serbia), located approximately 100 km SW of Belgrade (Figure 1).

Samples were collected at exactly the same place where they were collected previously in 1995 and 2010 [42,43]; a forest composed of mixed wood with two dominant species: horn-beam (*Carpinus betulus*) and oak (*Quercus robur*) together with other tree species (*Fraxinus angustifolia*, *Crataegus monogyna*, *Tilia argentea*, *Cornus mas*, *Acer campestre*) as well as brushwood plants (*Ruscus aculeatus*, *Asperula odorata*, *Hedera helix*, *Fragaria vesca*, *Prunella vulgaris*, *Carex silvatica*, and others). The flies were collected in 2019 (from 10 to 12 June), 2020 (from 13 to 17 June), 2021 (from 13 to 15 June), and 2022 (from 12 to 14 June). Each year, 40 plastic boxes filled with yeast-fermented apples were used as bait, and the flies were netted in the afternoon from 4 to 8 p.m. 

Once in the laboratory, *D. subobscura* wild males and sons of wild females were crossed in individual vials with virgin females from the Kussnacht reference strain. In all five chromosomes of the species (A, E, J, U, and O), this strain is homokaryotypic. All crosses were carried out under the same experimental conditions (18 °C, 60% relative humidity, and 12 h/12 h light/dark cycle). The following procedure was followed to obtain the chromosomal preparations: third instar larvae from the previous crosses were dissected and polytene chromosomes were stained and squashed in an aceto-orcein solution. For obtaining the karyotypes (the inversions for both homologous chromosomes from each pair) with a probability higher than 0.99, at least eight larvae from the progeny of each cross were examined. The chromosomal maps of Kunze-Mühl and Müller [47] and Krimbas [16,17] were used to accurately identify the inversions.

The Avala (Serbia) and Font Groga (Barcelona, Spain) populations were used with the aim of comparison. Avala (44°41′25′′ N 20°30′51′′ E) is a mount (450 m a.s.l.) located 18 km south of Belgrade. In the years 2014–2017, flies were sampled from a forest with polydominant communities of *Fagetum submontanum mixtum* [40]. Font Groga (41°25′54′′ N 2°07′20′′ E) is located in the foothills of Mount Tibidabo (415 m a.s.l.) on the edge of the Barcelona city limit. *D. subobscura* were collected during the years 2011–2015 in a forest of *Pinus pinea* and *Quercus ilex* [41].

### 2.2. Climatic Information

The meteorological data from Petnica; mean temperature (Tmean), maximum temperature (Tmax), minimum temperature (Tmin), mean humidity (Hm), and rainfall (Rf); were provided by the Serbian Republic Hydrometeorological Service. Temperature, humidity, and rainfall were expressed in degrees centigrade, percentage, and millimeters of precipitation, respectively. All populations used in this study were classified according to the Köppen–Geiger climate classification [38,39,48] using the climatologies at high resolution for the Earth’s land surface areas obtained through the CHELSA V2.1 software [49].

### 2.3. Statistical Analyses

The basic and vegan packages of R language [50] were used to carry out all statistical computations. To study the possible departure of the observed frequencies of chromosomal karyotypes from Hardy–Weinberg expectations and to compare the chromosomal composition between different years, Fisher’s exact test was used (statistically significant *p* < 0.05). Using a bootstrap procedure (100,000 runs), the corresponding *p* values were obtained. The FDR correction [51] was applied in all cases of multiple comparisons, and it was reported as significant for *p* < 0.05. The index of free recombination (IFR) and the chromosomal thermal index (CTI) were computed according to [52] and [53], respectively. A comparison between the Petnica (2019–2022), Avala (2014–2017), and Font Groga (2011–2015) populations using the inversion polymorphism from all chromosomes (A, J, U, E, and O) was carried out according to the following procedure. A principal coordinate analysis with this group of populations was computed using the Bhattacharyya distance [54]. Furthermore, a cluster was generated with the same data using the GEVA-Ward procedure, because for chromosomal inversion data it is considered very reliable [36,37]. Finally, the Pearson cophenetic correlation was computed to quantify how faithfully the cluster maintained the pairwise distances between the original data.

The relationship between climatic variables (mean, maximum, and minimum temperature, mean humidity, and rainfall) and chromosomal inversions was computed only for those inversions reported in all Petnica samples for the period (2019–2022). With this criterion, the infrequent inversions U_1_, O_7_, O_22_, O_3+4__+7_, and O_3+4__+17_ were not included in the analysis. Moreover, the meteorological information from March, April, and May were gathered together during the studied period (2019–2022), because it was assumed that the climate of these previous months before trapping the *D. subobscura* flies could have a large influence on inversion composition of the samples. The mean values of these climatic variables from these three months were used for the computations (Appendix A). To summarize the information of these climatic variables, a principal component analysis (PCA) was computed. Thus, the information provided by the climatic variables was synthesized using the principal components. A Poisson regression model using the first principal components as regressors was computed for each chromosome and inversion. The FDR correction [51] was applied for the individual significance of each component, and it was considered as significant for *p* < 0.05. Moreover, the ratio of the variation of the frequency of chromosomes when all climatic variables were kept constant and only one variable was either increased or decreased in only one unit was evaluated.

To assess the possible global warming effect in the Petnica population, a temporal series was computed using recorded data for the month of June (1995–2022) for the climatic variables of mean, maximum, and minimum temperature, mean humidity, and rainfall. Moreover, possible differences between frequencies of ‘cold’-, ‘warm’-, and ‘non-thermal’-adapted inversions for the Petnica samples from 1995 and the period 2019–2022, all collected in June, were analyzed using the previously explained Fisher’s exact test.

Finally, to ascertain whether the chromosomal inversion polymorphism for the O chromosome (the longest and most polymorphic) of different populations from the Balkan region was related to the Köppen–Geiger climate classification and to detect possible effects of global warming on them, a principal coordinate analysis (PCoA) and a GEVA-Ward cluster were computed using the same previously described procedure. The Balkan populations used were mainly from Serbia: Apatin [42,46], Avala [40], Djerdap [45], Jastrebac [44], Kamariste [42], Petnica (present research and [43]), Fruska Gora [55], and also the Montenegrin population of Zanjic [42] (Figure 1). In these studies, the Fruska Gora population was included for its interesting geographical location, although precise information of month and year of sampling is not provided in the paper. As outgroups, the Mt. Parnes (Greece) [56] and Font Groga (Barcelona, Spain) [41] populations were used. Moreover, to compare the CTI values of the samples belonging to the three climatic regions (Balkan Cfa, Balkan Cfb, and Barcelona Csa), a one-way ANOVA with Tukey post-hoc test was computed. For this CTI analysis, samples from the Observatori Fabra (also in Barcelona, Spain) population, with similar environmental and ecological conditions and located only 2 km from Font Groga, were also used. From this population, the November samples of the years 1987, 1988, 1989, and 2011 were utilized [33].

## 3. Results

### 3.1. The Chromosomal Inversion Polymorphism of the Petnica Population

The chromosomal inversion frequencies for Petnica during the studied years 2019–2022 are presented in Table 1.

The chromosomal inversion J_3+4_, typical of arid climates [16,17], was detected in all studied samples, although in low frequencies (around 1%). This inversion was not reported in the 1995 and 2010 samples from Petnica [43]. The U_1_ and O_7_ inversions, products of unlikely recombination events, were sporadically detected and in low frequencies. The first one was already reported in June 1995 in Petnica [43]. For the O chromosome, the inversions O_3+4__+7_ and O_3+4__+17_ were not detected in all years and were not observed in the 1995 and 2010 samples from Petnica either [43]. The O_3+4__+7_ is rather frequent in the Iberian Peninsula, Turkey, and American colonizing populations, whereas O_3+4__+17_ is reported in the Iberian Peninsula, but in very low frequencies [16,17]. Interestingly, U_1__+2+3_ was observed in Petnica (June 1995), but not in the present research samples. Previously, it was only reported in Avala (June 2017) [40]. In the present samples, the ‘warm’-adapted inversions U_1+8__+2_ and E_1+2+9+12_ were always observed, as they were in 2010, but not in 1995 [43]. During the years 2019–2022, the O_3+4__+6_ inversion was always observed in low frequencies, but previously it was only detected in the June 1995 collection from Petnica [43]. In Table 2, the karyotypes for Petnica’s samples (2019–2022) are shown. 

There was no deviation of the Hardy–Weinberg equilibrium in any sample, with the exception of the O chromosome for 2022 (Appendix A). This deviation could be explained by an excess of the homokaryotypes O_st_/O_st_ and O_3+4_/O_3+4_, and a lack of the heterokaryotypes O_st_/O_3+4_. However, this result could be spurious, because this effect was not observed in the other chromosomes (J, U, and E) of the same sample. During 2019–2022, the IFR values (Table 2) were high and rather similar, although a slight trend to increase was observed. These values would indicate that Petnica can be considered a population located in the central area of the species distribution. 

### 3.2. Comparison between the Inversion Chromosomal Polymorphism of Petnica, Avala, and Font Groga 

Although it is possible to qualitatively compare the composition of inversions (types and frequencies) between these three populations, it is better to compare them quantitatively by means of multivariate analyses (Figure 2). 

Regarding the PCoA, first, second, and third axes explained 77.55%, 6.94%, and 5.64%, respectively. Samples from the Petnica (Cfa: humid subtropical climate) and Avala (Cfb: temperate oceanic climate or subtropical highland climate) populations appeared clearly separated and those from Font Groga (Csa: hot-summer Mediterranean climate) were located further away (Figure 2A). This distribution was confirmed by the cluster study (Figure 2B), in which the first partition separated the Font Groga samples and those from the Balkans, and the second partition properly identified two groups (the Petnica and Avala samples). The accuracy of the tree was excellent because the cophenetic correlation coefficient was 0.97. 

### 3.3. O Chromosome Inversion Polymorphism from the Balkan Populations 

The similarity between the Balkan populations of *D. subobscura* based on their composition of inversions for the O chromosome was studied. As previously explained, the Mt. Parnes (Greece) and Font Groga (Barcelona, Spain) populations were included in the analysis as reference outgroups. The obtained cluster is shown in Figure 3. 

Its cophenetic correlation was 0.78, showing good accuracy of the tree. The first partition clearly separated the Font Groga samples from all others (from the top down in the tree, population numbers 34–36). The second partition differentiated two groups, the first mainly constituting the most recent samples from the Balkans (from the top down in the tree, population numbers 6–26), whereas the second constituted the remaining populations. Interestingly, in the group composed of population numbers 6–26, a pair of subgroups can be observed. The first one (population numbers 6–14) was composed of the samples from Avala (Cfb climate) and the second (population numbers 8–26) mainly of the populations present in the Cfa climate. The large group of remaining populations (from the top down in the tree, population numbers 30–23) was a mixture of samples from different locations, months, and climates (Cfa and Cfb), with Mt. Parnes (Greece) inside this group (number 1, Csb climate). Studying this group in particular, it can be observed that, generally, the last partitions correspond to the samples from the same population or from the same climate. In the PCoA analysis (Appendix A), first, second, and third axes explained 37.84%, 16.94%, and 9.93%, respectively. It showed similar results as those obtained from the cluster analysis, but more difficult to observe.

### 3.4. Relationship between Chromosomal Inversions and Meteorological Variables

In Table 3, the principal component analysis of climatic data from the Petnica population is presented. Using the climatic variables (Tmean, Tmax, Tmin, Hd, and Rf), the first three principal components (PC1, PC2, and PC3) accounted for 59.73%, 39.75%, and 0.52% of the variance, respectively. 

Thus, all three components explained the 100% of the variability and no information was lost. The first component (PC1) is mainly dependent on temperature (Tmean, Tmax, and Tmin) and partially on Rf; the second one (PC2) is related to Hm and Rf, and finally, the effects of all variables can be considered negligible for PC3. In Table 4, the individual significance of PC1, PC2, and PC3 for the inversions from Petnica is presented. 

After the FDR correction, only E_st_ (‘cold’-adapted) and E_1+2+9_ (‘warm’-adapted) remained significant and both for the PC2, a component related to Hm and Rf. Before the adjustment, A_1_ and J_1_ were significant for PC1 and PC2; U_1+2+6_ and E_1+2+9_ were significant for PC2 and PC3; and finally, O_3+4_ was only significant for PC1, E_st_ and E_8_ for PC2, and U_1+8__+2_ for PC3. Finally, in Table 5, when all variables were kept constant, but only one variable of interest changed by one unit, the rations of variation in the quantity of inversions for each chromosome are shown. 

In Petnica, the studied climatic variables generated, in most cases, small variations in the abundance of inversions. Only a few cases could be highlighted: E_1+2+9+12_ (‘warm’-adapted) showed an increase with temperature and a small decrease with Hm; O_6_ clearly increased with temperature, presented a small increase with Rf, and decreased with Hm; and finally, O_3+4__+6_ showed almost the opposite behavior than O_6_, decreasing with all studied variables.

### 3.5. Changes over Time in the Chromosomal Polymorphism from Petnica

For all chromosomes (A, J, U, E, and O), the comparisons between the inversion chromosomal polymorphism for 1995 and those from 2019, 2020, 2021, and 2022 are presented in Appendix A. For the comparison between the 1995 and 2019 samples, all chromosomes presented significant differences with the exception of A and O. For the second one (1995 and 2020), significant differences were reported for the U, E, and O chromosomes, although the significance was lost when FDR correction was applied for the O chromosome (adjusted *p* = 0.0560). In the case of the comparison between 1995 and 2021, only the U and O chromosomes presented significant differences, but the latter was lost with the FDR correction (adjusted *p* = 0.0560). Finally, when comparing the 1995 and 2022 inversion compositions, the U, E, and O chromosomes presented significant differences. However, considering the years 1995, 2019, 2020, 2021, and 2022, it is interesting to observe the changes over time in the ‘cold’—(Figure 4a), ‘warm’—(Figure 4b), and ‘non-thermal’—(Figure 4c) adapted inversions for the Petnica population. 

For the A chromosome, there was an increase in frequency for ‘cold’-adapted inversions (A_st_ and A_1_) and a decrease in that of ‘warm’-adapted inversions (A_2_). Regarding the J chromosome, the ‘cold’-adapted inversions (J_st_) increased in frequency a little bit from 1995 to 2019, and then it decreased to finally reach a similar frequency to that of 1995. For the ‘warm’-adapted (J_1_) frequency, the behavior over the years was just the opposite, and the fluctuations of the ‘non-thermal’-adapted (J_3+4_) frequency were irrelevant. For the U chromosome, the frequency of the ‘cold’-adapted (U_st_) remained rather constant. The ‘warm’-adapted inversions showed a clear increase in frequency, although an abrupt decrease was observed in 2022, whereas the opposite behavior was observed for the ‘non-thermal’-adapted inversions. In the case of the E chromosome, the ‘cold’-adapted inversions (E_st_) slowly increased in frequency until 2020, then a sharp drop took place in 2021, and finally, a pronounced increase was detected in 2022. The frequency of ‘warm’-adapted inversions showed approximately the opposite behavior over time. However, the frequency of the ‘non-thermal’-adapted inversions tended to increase slightly in the period 2019–2022. Finally, for the O chromosome, the frequency of the ‘cold’-adapted inversions (O_st_) remained rather constant from 1995 to 2022, with global a slight increase over time. A similar behavior was observed for the ‘warm’-adapted inversions, although a small increase in frequency was shown in 2020. Finally, the ‘non-thermal’-adapted inversions showed negligible frequency variations over time. 

Furthermore, for these three types of inversions (‘cold’-, ‘warm’-, and ‘non-thermal’-adapted), the statistical differences between the 1995 and 2019, 1995 and 2022, and finally, 2019 and 2022 samples are shown in Appendix A. For the ‘cold’-adapted inversions, significant differences after the FDR correction were observed in the J chromosome (1995 vs. 2019) and E chromosome (1995 vs. 2022). In the case of ‘warm’-adapted inversions, significant differences were detected for the J chromosome (1995 vs. 2019), the U chromosome (in all three comparisons), and E chromosome (1995 vs. 2022 and 2019 vs. 2022). Finally, regarding the ‘non-thermal’-adapted inversions, there were only significant differences for the U chromosome (in all three comparisons).

All these variations of the chromosomal polymorphism have to be related with the meteorological variables from the Petnica population during the period 1995–2022. Thus, the temporal series for the climatic variables used (Tmean, Tmax, Tmin, Hm, and Rf) presented annual variations (Appendix A). All temperatures showed an upward trend, with Tmean and Tmin being significant (*p* = 0.0163 and 0.0002, respectively) and Tmax was not significant (*p* = 0.0861). Humidity and rainfall followed rather erratic distributions, as expected in a global warming scenario.

### 3.6. CTI in the Balkan Populations of D. subobscura

It is interesting to observe the variation of CTI in the samples from Petnica for the month of June: 0.281 (1995) [43], 0.165 (2019), 0.170 (2020), 0.211 (2021), and 0.097 (2022) (Table 1). The first value is rather high, but is lower after 24 years (2019). Then, a constant increase is observed for the period 2019–2021, but a sharp decline is observed in 2022. The statistical comparisons between these CTI values are presented in Appendix A. To understand and give a proper explanation of this succession of CTI values, it is necessary to study in detail the climatic conditions of Petnica from 1994 (the year before to the first studied sample) to 2022.

All CTI values obtained in the Balkan populations are presented in Appendix A. The mean (±SD), maximum, and minimum estimates were 0.244 ± 0.120, 0.426, and −0.347, respectively. According to different climates, the mean, maximum, and minimum CTI values for the Balkan populations with a Cfa climate were 0.207 ± 0.094, 0.354, and −0.347, respectively. In the case of Balkan populations with a Cfb climate, the corresponding CTI estimates were 0.281 ± 0.130, 0.426, and −0.062. It is valuable to compare those results with the corresponding results from Barcelona (Font Groga and Observatori Fabra together), which is a location presenting another climate (Csa). In this case, the mean, maximum, and minimum values for the CTI were 0.427 ± 0.107, 0.604, and 0.285, respectively. The one-way ANOVA showed significant differences between these three groups (*p* = 0.0002). The Tukey post-hoc test indicated significant differences in the comparisons between Balkan Cfa and Balkan Cfb, and also between Balkan Cfa and Barcelona Csa (adjusted *p* = 0.0277 and 0.0002, respectively), but not for Balkan Cfb and Barcelona Csa (adjusted *p* = 0.0891).

## 4. Discussion

Global warming is a fact, and the temperatures in Petnica were in agreement with this phenomenon. All temperatures increased over time, where Tmean and Tmin were significant and Tmax was not significant. Similar results were reported in Avala [40] and Font Groga [41]. Thus, the main evolutionary question is, how is *D. subobscura* able to adapt to this environmental change? The inversion chromosomal polymorphism makes this species a magnificent model through which to study the effects of natural selection on its genetic material to achieve adaptations. The adaptive potential of this polymorphism was observed in this research, because the three analyzed populations (Petnica, Avala, and Font Groga) presenting different climates (Cfa, Cfb, and Csa, respectively) were clearly differentiated with regard to chromosomal inversions (Figure 2). Also, through revisiting data on the inversion chromosomal polymorphism from the Balkan region, although temporal changes should be considered, a clear differentiation of populations based on their climatic region was observed (Figure 3). Furthermore, focusing on the composition of ‘warm’- and ‘cold’-adapted inversions, these three populations (Petnica, Avala, and Font Groga) differed in the range of CTI values. The effect of selection is strong on this polymorphism because *D. subobscura* is a species with a high dispersion potential [57,58,59], a factor that, generating intense gene flow, tends to homogenize the genetic composition of populations. Obviously, this gene flow depends on the natural characteristics of the species and the facilitating human effect, and was reported in Palearctic and American colonizing populations [56,60]. Also, selection acted very fast in establishing latitudinal clines for the chromosomal inversion polymorphism in the American continent after the colonizing events of the South and North by *D. subobscura* [21,61]. Moreover, strong selection effects on chromosomal inversions were reported in other species of the *Drosophila* genus [62].

In Petnica, only E_st_ (‘cold’) and E_1+2+9_ (‘warm’) inversions were significant, and both for the PC2, a component related to humidity and rainfall (Table 4). However, in the Avala population [40], A_st_ (‘cold’) and the ‘warm’ inversions J_1_, U_1__+2_, E_1+2+9_, and O_3+4_ proved to be significant for PC1 and PC2, components related to temperature, humidity, and rainfall. In Font Groga [41], most inversions were significant for PC2, a component related to minimum temperature and negatively to maximum temperature. On the other hand, when analyzing the variation of only one variable in one unit in Petnica (Table 5), E_1+2+9+12_ (‘warm’) showed an increase with temperature and a small decrease with Hm. For the O_3+4__+6_ and O_6_ inversions, both ‘non-thermal’-adapted inversions showed opposite behaviors regarding temperature, humidity, and rainfall. In Avala, E_1+2+9+12_ and O_3+4__+6_ showed an effect in relation to climatic variables, the first one in the same sense as in Petnica, but the second one in the opposite direction [40]. In Font Groga, more inversions reacted to climatic variables [41]. As a general conclusion for these analyses, the chromosomal inversions would react differently under distinct climatic circumstances and locations.

As previously commented, temperatures changed significantly over time in Petnica, according to global warming expectations. However, the variations on chromosomal inversion composition and frequencies were particular. For instance, the 1995 sample presented a rather high value of CTI (0.281); it was low in 2019 (0.165) and increased along 2020 and 2021 (0.170 and 0.211, respectively), finally showing an abrupt decrease in 2022 (0.097) (Table 1), where the comparisons between 1995 vs. 2022 and 2021 vs. 2022 were significant (Appendix A). This pattern can be explained if the climatic conditions for those years are considered. Thus, in 1994 in Europe, there was an intense summer heat wave [63,64] that could have conditioned the accumulation of ‘warm’ (and a reduction of ‘cold’) inversions in Petnica for the 1995 sample. On the other hand, the 2021–2022 winter was extremely cold (data from the Serbian Republic Hydrometeorological Service), and would explain the low CTI value obtained in June 2022. 

Furthermore, it was possible to quantify the changes in chromosomal inversion frequencies from these samples from Petnica (Appendix A). The chromosomes that presented more significant changes were the U and E. Particularly, all comparisons between ‘warm’ and ‘non-thermal’ U chromosome inversions were significant. From Figure 4b,c, it was possible to observe that frequency variations in U ‘warm’-adapted inversions (mainly for U_1__+2_) corresponded to frequency fluctuations of ‘non-thermal’-adapted inversions (U_1+2+6_), but not for the frequency changes in the ‘cold’-adapted inversions (U_st_). A possible explanation could be that these inversions were not correctly defined with regard to thermal adaptation or that these fluctuations would be due to adaptations to other physical and/or biological variables. However, this result was not new, because a similar effect was reported in the Avala population [40]. Remarkably, U was the chromosome that showed more significant frequency changes over time in several Balkan populations [40,43,46].

All these results from Petnica deserve to be commented on in depth for their evolutionary consequences on species adaptation. The increase in temperature (Tmean, Tmax, and Tmin) is not constant and regular, because there are variations throughout the seasons and in different years. For this reason, analyzing a couple of samples for their chromosomal inversion composition could yield anomalous results regarding global warming if the time elapsed between them is not enough. However, if the elapsed time is long enough, the differences in temperature will be sufficiently important and for this reason the chromosomal polymorphism will already be different. Also, it is worth remembering that the inversion chromosomal polymorphism is not always changing in the same direction, because in temperate climates there is an alternance of cold and warm seasons. Seasonal changes in the chromosomal polymorphism have been reported in *D. subobscura* [16,32,45,65,66]. Seasonal changes in this polymorphism were described in other species of the *Drosophila* genus (for a revision see [67]). Due to seasonality, *D. subobscura* populations from temperate climates shown a chromosomal polymorphism with ‘warm’- and ‘cold’-adapted inversions in different proportions depending on the time of year. As a consequence of these climatic conditions, there were reports of rich chromosomal polymorphism in *D. subobscura* populations of temperate climates, as was observed in the Atlantic, Central Europe, Iberian Peninsula, Italy, and Balkans populations (for a revision see [16,17]). However, in cold marginal populations the chromosomal polymorphism would be poor, and ‘warm’ inversions would be in low frequency because they are counterselected, but would not disappear due to a certain migration level, generating a dynamic equilibrium between both evolutionary forces (selection and migration) [68,69]. On the contrary, in warm marginal populations, the opposite effect would happen. The *D. subobscura* observations were in agreement with this model. Looking at Western Europe along the SW-NE axis, in the populations from Morocco ‘cold’ inversions have a very low frequency [70,71] and those from Norway and Sweden contained a negligible frequency of ‘warm’ inversions [72,73].

This means that natural selection is acting in the inverse sense according to the season, and also in the ‘background’ it is slowly increasing the ‘warm’ and decreasing the ‘cold’ inversions, because in general, summer seasons are becoming hotter and winter seasons more temperate. However, another question arises; at which temporal period does natural selection act on chromosomal inversions? The inversions observed in *D. subobscura* from a given sample are the consequence of selection in a period with particular climatic conditions. In this research, we considered the mean values of climatic variables from the three previous months, as explained in the Section 2. However, the 2022 inversion results from Petnica seemed to be conditioned from the previous winter, and thus, our temporal choice is questionable. Under laboratory conditions (18 °C), a new generation of *D. subobscura* is obtained in approximately three weeks, but in temperate Palearctic natural conditions, four to six generations per year are assumed [58,74]. However, they are not distributed at regular intervals over the year, and for this species a pair of abundance peaks exists, one in spring and the second one in autumn [17,75]. *Drosophilid* collections carried out by different authors indicated that, during the winter season, *D. subobscura* individuals are almost absent and the species probably goes on diapause (for a revision see [16,17]), defined as a resting period of suspended growth or development, characterized by greatly reduced metabolic activity [76]. For this reason, no new generations arise until spring, when they follow one another at a good pace over time until the end of autumn. Good environmental conditions in spring and early autumn are the periods of maximum abundance for the species. It is likely that selection would favor the chromosomal inversions that allow adaptation under winter conditions, thus in early spring a large frequency of ‘cold’-adapted inversions would be expected. As spring temperatures increase and generations of *D. subobscura* follow each other over the months, natural selection can choose ‘warm’-adapted inversions, which would increase in frequency over time. Selection will again select ‘cold’-adapted inversions in the last generations during late autumn (progressive decrease in temperature). For these reasons, depending on the sample collection month and year, it is possible to obtain different compositions for the inversion chromosomal polymorphism. Obviously, this temporal pattern is dependent on the climatic conditions of the studied populations, as was demonstrated in this research when comparing Petnica (Cfa), Avala (Cfb), and Font Groga (Csa).

Finally, it is worth remembering that chromosomal inversions do not only adapt to climatic variables, but to the ecosystem conditions. Meteorological variables are important, but the biological composition is also fundamental. Both aspects, often intertwined, are key elements for the survival and reproduction of the organisms and they have to adapt to them. In a particular ecosystem, *D. subobscura* flies have to feed, avoid predators and parasites, breed, and be efficient to other fitness components. Natural selection will choose those inversions which genetic content allows to survive and produce the best-fitted flies. Thus, genetic and genomic approaches are needed to properly identify, map, and characterize the genes, located inside or linked to inversions, involved in adaptation to thermal and other environmental conditions [27,28,77,78,79,80]. Although this is true, the concept of the supergene introduced by Dobzhansky [9] is still useful and inversion chromosomal polymorphism continues to demonstrate that it is an excellent marker to monitor the adaptation of organisms to global warming. 

## 5. Conclusions

In summary, *D. subobscura* populations from different climatic regions (Cfa, Cfb, and Csa) presented significant differences regarding the chromosomal inversion composition. Furthermore, CTI values that measure the content in thermal-adapted (‘warm’ or ‘cold’) inversions, accurately reflected these populational differences. In Petnica, significant global warming was reported, and changes in temperature in the annual series (1995 and 2019–2022) were related to thermal-adapted inversions, mainly from the U and E chromosomes. Moreover, the effect of the U chromosome was also reported in other Balkan populations, and its frequency variations over time for ‘warm’-adapted inversions corresponded with opposite changes in ‘non-thermal’ inversions. Also, it was possible to observe that the abundance of particular inversions from Petnica were related to different temperatures (mean, maximum, and minimum), but also to humidity and rainfall. 

## Figures and Tables

**Figure 1 insects-14-00596-f001:**
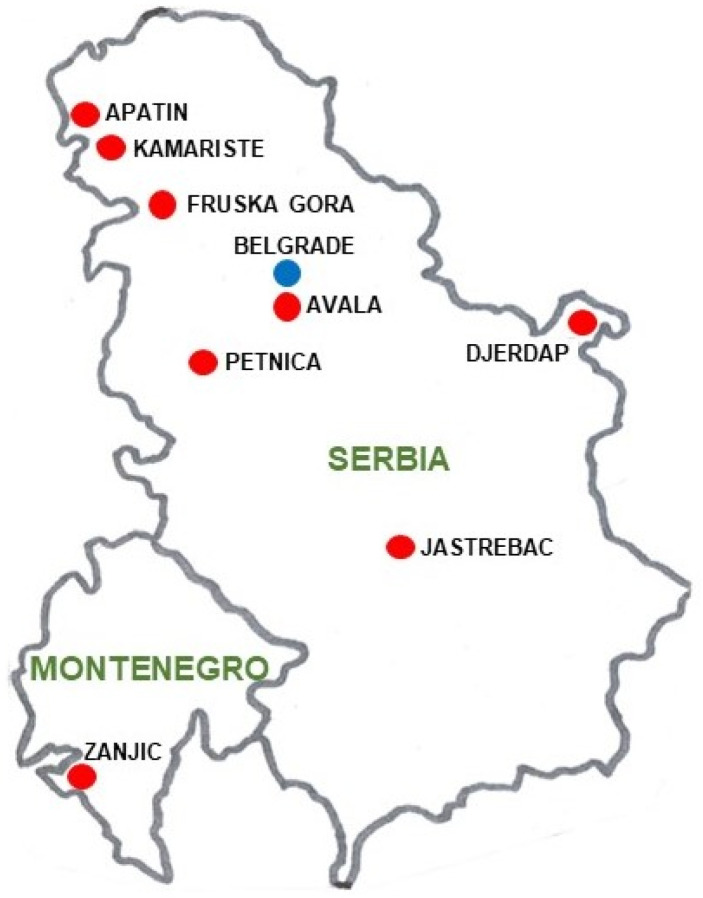
Map with the geographical location of the Serbian and Montenegrin *D. subobscura* populations used. Belgrade is also presented for geographical reference.

**Figure 2 insects-14-00596-f002:**
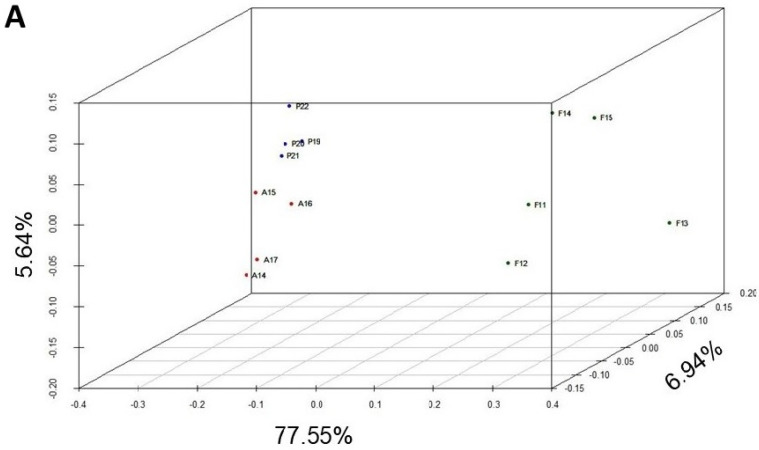
Multivariate analysis of the chromosomal inversion composition of the Petnica, Avala, and Font Groga populations of *D. subobscura*. (**A**) Principal coordinate analysis. Two groups are clearly differentiated on the left, corresponding to the Petnica and Avala collections. On the right is the group of samples from Font Groga. (**B**) GEVA-Ward cluster study. The first partition differentiates the Font Groga collections from those of the Serbian populations (Petnica and Avala). The second partition generates two groups, one with the Petnica samples and the other with those from Avala. The abbreviations used are: in blue, P19 (Petnica 2019), P20 (Petnica 2020), P21 (Petnica 2021), and P22 (Petnica 2022); in red, A14 (Avala 2014), A15 (Avala 2015), A16 (Avala 2016), and A17 (Avala 2017); in dark green, F11 (Font Groga 2011), F12 (Font Groga 2012), F13 (Font Groga 2013), F14 (Font Groga 2014), and F15 (Font Groga 2015).

**Figure 3 insects-14-00596-f003:**
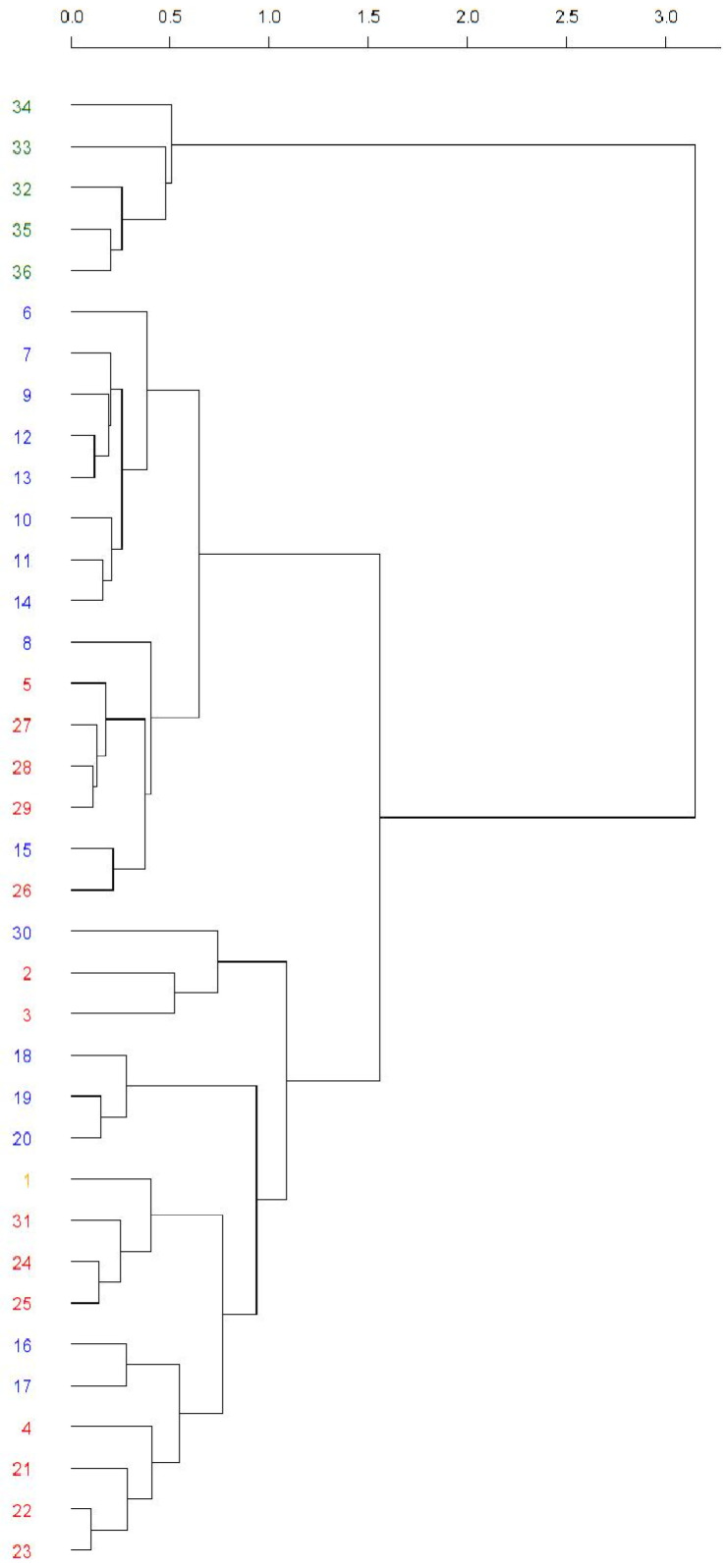
Cluster analysis of O chromosome inversion polymorphism from the Balkan populations. The Mt. Parnes (Grece) and Font Groga (Barcelona, Spain) populations were used as reference outgroups. The colors indicate the climate: dark green (Csa), blue (Cfb), red (Cfa), and orange (Csb). The numbers stand for the used populations: 1. Mt. Parnes May 2006; 2. Apatin June 1994; 3. Apatin June 2008; 4. Apatin June 2009; 5. Apatin June 2018; 6. Avala Sept. 2003; 7. Avala June 2004; 8. Avala Sept. 2004; 9. Avala Sept. 2005; 10. Avala June 2011; 11. Avala June 2014; 12. Avala June 2015; 13. Avala June 2016; 14. Avala June 2017; 15. Djerdap June 2001; 16. Djerdap Aug. 2001; 17. Djerdap June 2002; 18. Jastrebac June 1990; 19. Jastrebac June 1993; 20. Jastrebac June 1994; 21. Kamariste June 1996; 22. Petnica May 1995; 23. Petnica June 1995; 24. Petnica Aug. 1995; 25. Petnica May 2010; 26. Petnica June 2019; 27. Petnica June 2020; 28. Petnica June 2021; 29. Petnica June 2022; 30. Fruska Gora 1971? (year not specified in the paper); 31. Zanjic June 1997; 32. Font Groga Oct.–Nov. 2011; 33. Font Groga Oct.–Nov. 2012; 34. Font Groga Oct.–Nov. 2013; 35. Font Groga Oct.–Nov. 2014; 36. Font Groga Oct.–Nov. 2015. References for these populations are presented in the text.

**Figure 4 insects-14-00596-f004:**
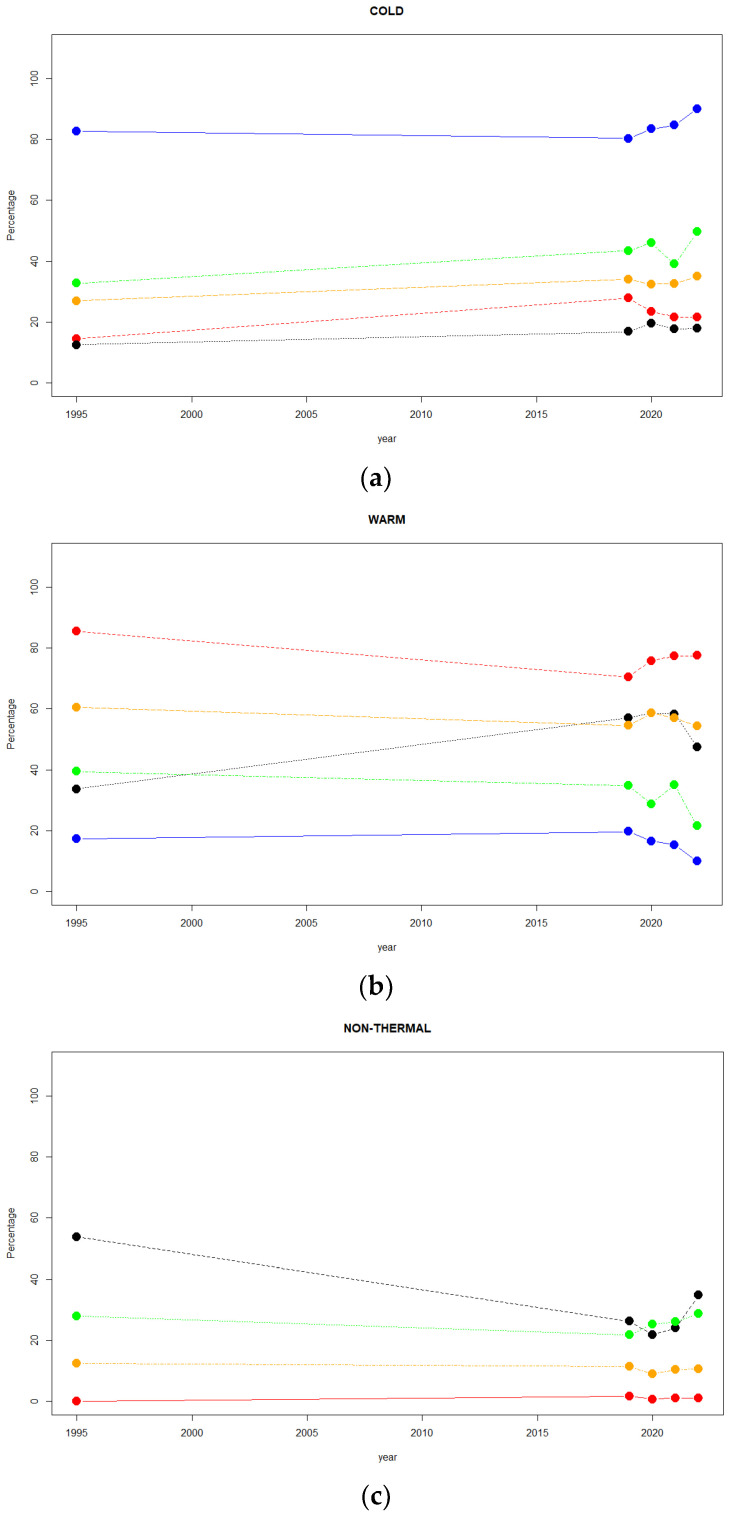
Changes in frequency (percentage) over the years (1995, 2019, 2020, 2021, and 2022) for the Petnica chromosomal inversions classified according to thermal adaptation. (**a**) ‘Cold’-adapted. (**b**) ‘Warm’-adapted. (**c**) ‘Non-thermal’. The filled circles stand for: blue for the A chromosome, red for the J chromosome, black for the U chromosome, green for the E chromosome, and orange for the O chromosome.

**Table 1 insects-14-00596-t001:** Chromosomal inversion frequencies of *D. subobscura* from the Petnica (Serbia) population in the consecutive years 2019–2022. Regarding thermal adaptation, ‘C’, ‘W’, and ‘N’ stand for ‘cold’-, ‘warm’-, and ‘non-thermal’-adapted, respectively. In the last row, CTI (Chromosomal Thermal Index) values are presented for each yearly sample.

Chrom. Inversion	Thermal Adapt.	Year			
	C/W	2019	2020	2021	2022
A_st_	C	54 (44.3%)	57 (37.7%)	61 (40.7%)	63 (41.7%)
A_1_	C	44 (36.1%)	69 (45.7%)	66 (44.0%)	73 (48.3%)
A_2_	W	24 (19.6%)	25 (16.5%)	23 (15.3%)	15 (9.9%)
TOTAL		122	151	150	151
J_st_	C	68 (27.8%)	71 (23.5%)	65 (21.7%)	65 (21.5%)
J_1_	W	172 (70.5%)	229 (75.8%)	232 (77.3%)	234 (77.5%)
J_3+4_	N	4 (1.6%)	2 (0.7%)	3 (1.0%)	3 (1.0%)
TOTAL		244	302	300	302
U_st_	C	41 (16.8%)	59 (19.5%)	53 (17.7%)	54 (17.9%)
U_1_	N	0	3 (0.9%)	2 (0.7%)	3 (1.0%)
U_1__+2_	W	132 (54.1%)	160 (52.9%)	164 (54.7%)	138 (45.7%)
U_1+2+6_	N	64 (26.2%)	63 (20.9%)	70 (23.3%)	102 (33.8%)
U_1+8__+2_	W	7 (2.9%)	17 (5.6%)	11 (3.7%)	5 (1.6%)
TOTAL		244	302	300	302
E_st_	C	106 (43.4%)	139 (46.0%)	117 (39.0%)	150 (49.7%)
E_8_	N	43 (17.6%)	67 (22.2%)	62 (20.7%)	75 (24.8%)
E_1+2_	N	10 (4.1%)	9 (2.9%)	16 (5.3%)	12 (3.9%)
E_1+2+9_	W	70 (28.7%)	77 (25.5%)	98 (32.7%)	50 (16.5%)
E_1+2+9+12_	W	15 (6.1%)	10 (3.3%)	7 (2.3%)	15 (4.9%)
TOTAL		244	302	300	302
O_st_	C	83 (34.0%)	98 (32.4%)	98 (32.7%)	106 (35.1%)
O_6_	N	6 (2.4%)	4 (1.3%)	1 (0.3%)	4 (1.3%)
O_7_	N	4 (1.6%)	0	0	0
O_3+4_	W	88 (36.1%)	131 (43.4%)	126 (42.0%)	115 (38.1%)
O_3+4__+1_	W	35 (14.3%)	35 (11.6%)	36 (12.0%)	32 (10.6%)
O_3+4__+6_	N	2 (0.8%)	7 (2.3%)	6 (2.0%)	8 (2.6%)
O_3+4__+7_	N	1 (0.4%)	0	3 (1.0%)	1 (0.3%)
O_3+4+8_	W	10 (4.1%)	11 (3.6%)	9 (3.0%)	17 (5.6%)
O_3+4__+17_	N	0	1 (0.3%)	0	0
O_3+4__+22_	N	15 (6.1%)	15 (4.9%)	21 (7.0%)	19 (6.3%)
TOTAL		244	302	300	302
CTI		0.165	0.170	0.211	0.097

**Table 2 insects-14-00596-t002:** Chromosomal karyotype frequencies of *D. subobscura* from the Petnica population in the consecutive years 2019–2022. In the last row, IFR stand for Index of Free Recombination.

	Year			
Chromosomal Karyotypes	2019	2020	2021	2022
J_st_/J_st_	12 (9.8%)	12 (7.9%)	8 (5.3%)	12 (7.9%)
J_st_/J_1_	44 (36.1%)	47 (31.1%)	48 (32.0%)	41 (27.1%)
J_st_/J_3+4_	0	0	1 (0.7%)	0 (%)
J_1_/J_1_	62 (50.8%)	90 (59.6%)	91 (60.7%)	95 (62.9%)
J_1_/J_3+4_	4 (3.3%)	2 (1.3%)	2 (1.3%)	3 (1.9%)
TOTAL	122	151	150	151
U_st_/U_st_	3 (2.4%)	11 (7.3%)	10 (6.7%)	6 (3.9%)
U_st_/U_1_	0	0	1 (0.7%)	0
U_st_/U_1__+2_	21 (17.2%)	19 (12.6%)	26 (17.3%)	22 (14.6%)
U_st_/U_1+2+6_	13 (10.6%)	10 (6.6%)	6 (4.0%)	20 (13.2%)
U_st_/U_1+8__+2_	1 (0.8%)	8 (5.3%)	0	0
U_1_/U_1_	0	1 (0.7%)	0	1 (0.7%)
U_1_/U_1__+2_	0	0	1 (0.7%)	0
U_1_/U_1+2+6_	0	1 (0.7%)	0	1 (0.7%)
U_1__+2_/U_1__+2_	36 (29.5%)	50 (33.1%)	46 (30.7%)	43 (28.5)
U_1__+2_/U_1+2+6_	34 (27.8%)	36 (23.8%)	36 (24.0%)	26 (17.2%)
U_1__+2_/U_1+8__+2_	5 (4.1%)	5 (3.3%)	9 (6.5%)	4 (2.6%)
U_1+2+6_/U_1+2+6_	8 (6.5%)	6 (3.9%)	14 (9.3%)	27 (17.9%)
U_1+2+6_/U_1+8__+2_	1 (0.8%)	4 (2.6%)	0	1 (0.7%)
U_1+8__+2_/U_1+8__+2_	0	0	1 (0.7%)	0
TOTAL	122	151	150	151
E_st_/E_st_	21 (17.2%)	42 (27.8%)	32 (21.3%)	39 (25.8%)
E_st_/E_1+2_	7 (5.7%)	5 (3.3%)	4 (2.7%)	3 (1.9%)
E_st_/E_1+2+9_	27 (22.1%)	23 (15.2%)	33 (22.0%)	24 (15.9%)
E_st_/E_1+2+9+12_	9 (7.4%)	4 (2.6%)	0	9 (5.9%)
E_st_/E_8_	21 (17.2%)	23 (15.2%)	16 (10.7%)	36 (23.8%)
E_1+2_/E_1+2_	0	0	2 (1.3%)	0
E_1+2_/E_1+2+9_	1 (0.8%)	1 (0.7%)	4 (2.7%)	2 (1.3%)
E_1+2_/E_1+2+9+12_	0	0	1 (0.7%)	1 (0.7%)
E_1+2_/E_8_	2 (1.6%)	3 (2.0%)	3 (2.0%)	6 (3.9%)
E_1+2+9_/E_1+2+9_	15 (12.3%)	14 (9.3%)	19 (12.7%)	7 (4.6%)
E_1+2+9_/E_1+2+9+12_	1 (0.8%)	0	2 (1.3%)	1 (0.7%)
E_1+2+9_/E_8_	11 (9.0%)	25 (16.5%)	21 (14.0%)	9 (5.9%)
E_1+2+9+12_/E_1+2+9+12_	2 (1.6%)	1 (0.7%)	0	2 (1.3%)
E_1+2+9+12_/E_8_	1 (0.8%)	4 (2.6%)	4 (2.7%)	0
E_8_/E_8_	4 (3.3%)	6 (3.9%)	9 (6%)	12 (7.9%)
TOTAL	122	151	150	151
O_st_/O_st_	12 (9.8%)	17 (11.2%)	16 (10.7%)	36 (23.8%)
O_st_/O_6_	1 (0.8%)	3 (2%)	1 (0.7%)	2 (1.3%)
O_st_/O_7_	4 (3.3%)	0	0	0
O_st_/O_3+4_	30 (24.6%)	43 (28.5%)	41 (27.3%)	20 (13.2%)
O_st_/O_3+4__+1_	14 (11.5%)	11 (7.3%)	13 (8.7%)	6 (3.9%)
O_st_/O_3+4__+6_	0	1 (0.7%)	3 (2.0%)	0
O_st_/O_3+4__+7_	1 (0.8%)	0	0	1 (0.7%)
O_st_/O_3+4+8_	4 (3.3%)	1 (0.7%)	2 (1.3%)	3 (2.0%)
O_st_/O_3+4__+22_	5 (4.1%)	5 (3.3%)	6 (4.0%)	2 (1.3%)
O_6_/O_3+4_	3 (2.4%)	1 (0.7%)	0	0
O_6_/O_3+4+8_	0	0	0	2 (1.3%)
O_6_/O_3+4__+22_	2 (1.6%)	0	0	0
O_3+4_/O_3+4_	18 (14.7%)	28 (18.5%)	31 (20.7%)	31 (20.5%)
O_3+4_/O_3+4__+1_	9 (7.4%)	15 (9.9%)	12 (8.0%)	13 (8.6%)
O_3+4_/O_3+4__+6_	1 (0.8%)	3 (2.0%)	3 (2.0%)	3 (2.0%)
O_3+4_/O_3+4+8_	4 (3.3%)	9 (5.9%)	0	3 (2.0%)
O_3+4_/O_3+4__+17_	0	1 (0.7%)	0	0
O_3+4_/O_3+4__+22_	5 (4.1%)	3 (2.0%)	8 (5.3%)	14 (9.3%)
O_3+4__+1_/O_3+4__+1_	4 (3.3%)	1 (0.7%)	5 (3.3%)	5 (3.3%)
O_3+4__+1_/O_3+4__+6_	0	3 (2.0%)	0	0
O_3+4__+1_/O_3+4+8_	1 (0.8%)	0	1 (0.7%)	1 (0.7%)
O_3+4__+1_/O_3+4__+22_	3 (2.4%)	4 (2.6%)	0	2 (1.3%)
O_3+4__+6_/O_3+4__+6_	0	0	0	2 (1.3%)
O_3+4__+6_/O_3+4+8_	1 (0.8%)	0	0	0
O_3+4__+6_/O_3+4__+22_	0	0	0	1 (0.7%)
O_3+4__+7_/O_3+4__+7_	0	0	1 (0.7%)	0
O_3+4__+7_/O_3+4+8_	0	0	1 (0.7%)	0
O_3+4+8_/O_3+4+8_	0	0	2 (1.3%)	4 (2.6%)
O_3+4+8_/O_3+4__+22_	0	1 (0.7%)	1 (0.7%)	0
O_3+4__+22_/O_3+4__+22_	0	1 (0.7%)	3 (2.0%)	0
TOTAL	122	151	150	151
IFR	79.84 ± 0.90	82.28 ± 0.77	82.93 ± 0.75	84.45 ± 0.75

**Table 3 insects-14-00596-t003:** Principal component analysis of climatic data (after standardization) from the Petnica population.

Climatic Variables	Component Coefficients		
	PC1	PC2	PC3
Tmean	0.856	−0.205	−0.022
Tmin	0.829	0.274	0.108
Tmax	0.781	−0.405	−0.034
Rainfall	0.520	0.705	−0.083
Humidity	−0.117	0.872	0.012

**Table 4 insects-14-00596-t004:** Individual significance of PC1, PC2, and PC3 (before and after the FDR correction) for the chromosomal inversions from Petnica. Significant values (*p* < 0.05) are presented in bold.

Chrom.	Nominal *p*			Adjusted *p*		
	PC1	PC2	PC3	PC1	PC2	PC3
A_st_	0.4770	0.6510	0.7280	0.6559	0.7723	0.8712
A_1_	**0.0369**	**0.0336**	0.5584	0.2706	0.1379	0.8388
A_2_	0.7240	0.2220	0.1850	0.8424	0.4440	0.6783
Jst	0.7460	0.9680	0.6100	0.8424	0.9690	0.8388
J_1_	**0.0031**	**0.0376**	0.4429	0.0678	0.1379	0.8388
J_3+4_	0.7658	0.6590	0.5542	0.8424	0.7723	0.8388
U_st_	0.2470	0.2440	0.3170	0.4528	0.4473	0.8388
U_1__+2_	0.0924	0.7948	0.1344	0.3104	0.8743	0.5914
U_1+2+6_	0.3961	**0.0197**	**0.0229**	0.6224	0.1083	0.2519
U_1+8__+2_	0.5431	0.6232	**0.0065**	0.7028	0.7723	0.1430
E_st_	0.4403	**0.0029**	0.7555	0.6458	**0.0399**	0.8712
E_8_	0.0599	**0.0120**	0.7410	0.3065	0.0880	0.8712
E_1+2_	0.2100	0.6230	0.5210	0.4528	0.7723	0.8388
E_1+2+9_	0.0697	**0.0036**	**0.0466**	0.3065	**0.0399**	0.3420
E_1+2+9+12_	0.1230	0.3820	0.3240	0.3382	0.6465	0.8388
O_st_	0.2460	0.1990	0.9530	0.4528	0.4440	0.9566
O_6_	0.0988	0.4382	0.9566	0.3104	0.6886	0.9566
O_3+4_	**0.0150**	0.2150	0.0752	0.1650	0.4440	0.4136
O_3+4__+1_	0.9410	0.6670	0.7920	0.9410	0.7723	0.8712
O_3+4__+6_	0.1760	0.1080	0.5910	0.4302	0.3394	0.8388
O_3+4+8_	0.9030	0.1320	0.4710	0.9410	0.3630	0.8388
O_3+4__+22_	0.2810	0.9690	0.5660	0.4755	0.9690	0.8388

**Table 5 insects-14-00596-t005:** The ratio when increasing or decreasing only one unit for Tmean, Tmax, Tmin, Hm, and Rf and maintaining the other variables constant for the chromosomal inversions from Petnica.

Chrom.	Ratio Increasing 1 Unit					Ratio Decreasing 1 Unit				
	Tmean	Tmin	Tmax	Hm	Rfl	Tmean	Tmin	Tmax	Hm	Rf
A_st_	0.97	0.95	0.98	0.98	0.96	1.03	1.05	1.02	1.02	1.04
A_1_	0.91	0.86	0.94	0.90	0.84	1.10	1.17	1.06	1.11	1.19
A_2_	1.00	1.09	0.97	1.13	1.11	1.00	0.92	1.03	0.88	0.90
J_st_	1.02	1.02	1.02	1.00	1.01	0.98	0.98	0.98	1.00	0.99
J_1_	0.93	0.90	0.95	0.95	0.90	1.08	1.11	1.06	1.05	1.11
J_3+4_	1.05	1.09	1.02	1.11	1.16	0.95	0.92	0.98	0.90	0.86
U_st_	0.94	0.92	0.96	0.94	0.90	1.06	1.09	1.04	1.06	1.12
U_1__+2_	0.94	0.95	0.94	1.02	0.97	1.07	1.05	1.06	0.98	1.03
U_1+2+6_	0.99	0.91	1.02	0.89	0.90	1.01	1.10	0.98	1.12	1.12
U_1+8__+2_	0.89	0.99	0.87	1.11	0.98	1.13	1.01	1.15	0.90	1.02
E_st_	1.00	0.94	1.03	0.89	0.89	1.00	1.07	0.98	1.12	1.12
E_8_	0.92	0.86	0.96	0.88	0.83	1.08	1.17	1.04	1.13	1.20
E_1+2_	0.85	0.87	0.85	1.09	0.97	1.18	1.15	1.18	0.92	1.04
E_1+2+9_	0.88	0.98	0.85	1.19	1.07	1.14	1.02	1.17	0.84	0.93
E_1+2+9+12_	1.28	1.17	1.29	0.86	1.05	0.78	0.85	0.77	1.16	0.95
O_st_	0.96	0.93	0.98	0.95	0.92	1.04	1.07	1.02	1.05	1.08
O_6_	1.82	1.59	1.83	0.74	1.17	0.55	0.63	0.55	1.35	0.86
O_3+4_	0.91	0.9	0.93	0.96	0.89	1.10	1.11	1.08	1.04	1.12
O_3+4__+1_	0.99	1.01	0.98	1.03	1.02	1.01	0.99	1.02	0.97	0.98
O_3+4__+6_	0.77	0.66	0.86	0.75	0.62	1.30	1.52	1.17	1.33	1.62
O_3+4+8_	1.07	0.95	1.12	0.82	0.87	0.94	1.06	0.90	1.22	1.15
O_3+4__+22_	0.89	0.89	0.9	1.02	0.94	1.12	1.12	1.11	0.98	1.06

## Data Availability

All data of this research are presented in this article and available to readers.

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
