# Peer review of "The Adaptive Value of Chromosomal Inversions and Climatic Change—Studies on the Natural Populations of Drosophila subobscura from the Balkans"

_insects, 2023, doi:10.3390/insects14070596_

Round 1
Reviewer 1 Report
In this study the authors record the inversion polymorphism of a Serbian population of Drosophila subobscura for four subsequent years. The results are compared with previous data (years 1995 and 2010) from the same region as well as with data from another region of Serbia and a region in Spain, which present different climates. They also test the relation of the frequency of specific inversions with meteorological variables trying to show the adaptive value of inversion polymorphism to climate changes.
This is a neatly written paper, methods and results are presented in an appropriate way and conclusions as well as limitations are properly discussed. While not necessarily novel, conceptually, or methodologically, the data presented add to the understanding of genome adaptation to climate change and I believe that it deserves publication after minor revisions.
Below are some suggestions for these revisions:
· Paragraph 3.3, Lines 274-313, Figure 3 and S1: It is not clear to me what the purpose of this analysis is. Although it is presented in detail in the Results, it is completely ignored in the Discussion section. Maybe the authors should explain better its purpose and discuss its value or choose to omit it.
· Paragraph 3.4, Lines 316-317: “…accounted for 59.73%, 39.75% and 0.52% of the variance, respectively (Table 3)”. Table 3 does not contain this information. Please indicate where it is presented.
· Paragraph 3.4, Lines 322-325: “Thus, all three components explained the 100% of the variability and no information 322 was lost. The first component (PC1) is mainly depended on temperature (Tmean, Tmax 323 and Tmin) and partially on Rf; the second one (PC2) is related with Hm and Rf; finally, 324 the effects of all variables can be considered negligible for PC3” I think this is where Table 3 should be cited.
· Paragraph 3.4, Lines 336: “…A1 and J1 were significant for PC1” Please add “and PC2”
· Paragraph 3.5, Lines 359: “were reported for the J and E and O chromosomes” Please replace “J” with “U”.
· Paragraph 3.5, Lines 378: “…increase in frequency for of ‘cold’ adapted…” Please delete “of”.
· Paragraph 3.5, Lines 409-411: “All temperatures showed an upward trend, being Tmean and Tmax significant (p = 0.0163 and 0.0002, respectively) and Tmin was not significant (p = 0.0861). According to Figure S2 Tmin was significant and Tmax was not significant. Please correct. Same at Discussion lines 436-437.
· Paragraph 3.6, Lines 415: “…0.281 (1995)” where is this data presented?
“…0.165 (2019), 0.170 (2020), 0.211 (2021) and 0.097 (2022)” Please add (Table 1).
· Paragraph 3.6, Lines 417: Correct “2010” to “2019”
· Discussion lines 484-485: “Figures 2b and 2c” Correct to “Figures 4b and 4c”
· Supplementary data, Table S6: It is presented twice. Please delete the second table.
Author Response
REVIEWER 1:
In this study the authors record the inversion polymorphism of a Serbian population of Drosophila subobscura for four subsequent years. The results are compared with previous data (years 1995 and 2010) from the same region as well as with data from another region of Serbia and a region in Spain, which present different climates. They also test the relation of the frequency of specific inversions with meteorological variables trying to show the adaptive value of inversion polymorphism to climate changes.
This is a neatly written paper, methods and results are presented in an appropriate way and conclusions as well as limitations are properly discussed. While not necessarily novel, conceptually, or methodologically, the data presented add to the understanding of genome adaptation to climate change and I believe that it deserves publication after minor revisions.
Thank you very much.
We have improved the manuscript following your comments and suggestions. We are very grateful.
Below are some suggestions for these revisions:
Paragraph 3.3, Lines 274-313, Figure 3 and S1: It is not clear to me what the purpose of this analysis is. Although it is presented in detail in the Results, it is completely ignored in the Discussion section. Maybe the authors should explain better its purpose and discuss its value or choose to omit it.
We agree with the reviewer that this point has to be improved. We consider that results from Figure 3 are very valuable to understand the adaptive value of inversions. For this reason, in the new version of the manuscript, this figure is commented in the Discussion.
Paragraph 3.4, Lines 316-317: “…accounted for 59.73%, 39.75% and 0.52% of the variance, respectively (Table 3)”. Table 3 does not contain this information. Please indicate where it is presented.
We agree that this sentence was not enough clear. We have improved it in the new version of the manuscript.
Paragraph 3.4, Lines 322-325: “Thus, all three components explained the 100% of the variability and no information 322 was lost. The first component (PC1) is mainly depended on temperature (Tmean, Tmax 323 and Tmin) and partially on Rf; the second one (PC2) is related with Hm and Rf; finally, 324 the effects of all variables can be considered negligible for PC3” I think this is where Table 3 should be cited.
With the new sentence at the beginning of the paragraph, Table 3 is cited earlier, so potential readers can understand the general meaning of the paragraph.
Paragraph 3.4, Lines 336: “…A1 and J1 were significant for PC1” Please add “and PC2”
We have corrected the manuscript according to the reviewer’s comment.
Paragraph 3.5, Lines 359: “were reported for the J and E and O chromosomes” Please replace “J” with “U”.
It has been corrected.
Paragraph 3.5, Lines 378: “…increase in frequency for of ‘cold’ adapted…” Please delete “of”.
We have corrected this mistake.
Paragraph 3.5, Lines 409-411: “All temperatures showed an upward trend, being Tmean and Tmax significant (p = 0.0163 and 0.0002, respectively) and Tmin was not significant (p = 0.0861). According to Figure S2 Tmin was significant and Tmax was not significant. Please correct. Same at Discussion lines 436-437.
It is true. It was a mistake that has been corrected in Results and Discussion sections.
Paragraph 3.6, Lines 415: “…0.281 (1995)” where is this data presented?
“…0.165 (2019), 0.170 (2020), 0.211 (2021) and 0.097 (2022)” Please add (Table 1).
We agree with the reviewer. In the new version, we have indicated where to find this information.
Paragraph 3.6, Lines 417: Correct “2010” to “2019”
It was a mistake that has been corrected.
Discussion lines 484-485: “Figures 2b and 2c” Correct to “Figures 4b and 4c”
It has been corrected.
Supplementary data, Table S6: It is presented twice. Please delete the second table.
We agree with the reviewer. This “duplicate” has been deleted.
Reviewer 2 Report
Dear Authors,
I read your interesting study on the adaptive value of chromosomal inversions during climate changes in natural populations of Drosophila subobscura from the Balkans. I recommend minor Language editing and I also have some specific comments and recommendations for the improvement of the manuscript, which are stated below:
Line 16: »Many species try to survive by adapting to the new climatic conditions.« This sentence implies that adaptation is a concious process (following Lamarck's theory). Please reform this sentence in a way that you consider Darwin's Theory of Evolution.
Line 57: you can omit the words: was considered and still
Line 77: you may omit the words: For instance, and
Line 252: Figure 2 is of low resolution and is difficult to read. Also A) and B) are missing on each part of the figure. These should be improved.
Line 294: »Symbol “?” stand for: month and year information is absent in the paper.« There is no ? symbol in the figure – please correct either in the figure or in the text.
Figure 4: The presentation of the data is unclear – I suggest you use boxplot histogram for this kind of data presentation. You can plot all the data into one figure instead of separated figures for each climate adaptation.
I recommend minor Language editing.
Author Response
REVIEWER 2:
Dear Authors,
I read your interesting study on the adaptive value of chromosomal inversions during climate changes in natural populations of Drosophila subobscura from the Balkans.
Thank you very much.
We are very grateful for your comments and suggestions to improve the manuscript.
I recommend minor Language editing and I also have some specific comments and recommendations for the improvement of the manuscript, which are stated below:
Line 16: »Many species try to survive by adapting to the new climatic conditions.« This sentence implies that adaptation is a conscious process (following Lamarck's theory). Please reform this sentence in a way that you consider Darwin's Theory of Evolution.
Obviously, we were thinking in adaptation by means of natural selection. We have improved the sentence to avoid misinterpretation.
Line 57: you can omit the words: was considered and still
We agree with the reviewer. The words “was considered and still” have been deleted.
Line 77: you may omit the words: For instance, and
Following the reviewer’s recommendation, the words “For instance, and” have been deleted.
Line 252: Figure 2 is of low resolution and is difficult to read. Also A) and B) are missing on each part of the figure. These should be improved.
According to reviewer suggestion, we have improved Figure 2. Also, A) and B) have been added.
Line 294: »Symbol “?” stand for: month and year information is absent in the paper.« There is no ? symbol in the figure – please correct either in the figure or in the text.
In this Figure, the symbol “?” appears in the legend: “30. Fruska Gora 1971?”. This is an interesting population for the study, but the sampling year is not specified in the article. Most probably, sampling took place in 1971. We have modified the text of the Figure legend to improve its clarity.
Figure 4: The presentation of the data is unclear – I suggest you use boxplot histogram for this kind of data presentation. You can plot all the data into one figure instead of separated figures for each climate adaptation.
We have experience from the previous researches (Galludo et al. 2018; Zivanovic et al. 2021) and we consider that the best representation is the current Figure 4. Trying to put boxplot histograms and all information in only one figure is really difficult to interpret. For these reasons, we have no changed Figure 4.
Comments on the Quality of English Language
I recommend minor Language editing.
Minor language mistakes have been checked and corrected.